# A Computational Study on the Mechanism of Catalytic Cyclopropanation Reaction with Cobalt N-Confused Porphyrin: The Effects of Inner Carbon and Intramolecular Axial Ligand

**DOI:** 10.3390/molecules27217266

**Published:** 2022-10-26

**Authors:** Osamu Iwanaga, Mayuko Miyanishi, Toshihiro Tachibana, Takaaki Miyazaki, Yoshihito Shiota, Kazunari Yoshizawa, Hiroyuki Furuta

**Affiliations:** 1Department of Applied Chemistry, Graduate School of Engineering, Fukuoka 819-0395, Japan; 2Institute for Materials Chemistry and Engineering, Kyushu University, Fukuoka 819-0395, Japan; 3Department of Chemistry, Faculty of Science, Fukuoka University, Fukuoka 814-0180, Japan; 4Research Organization of Science and Technology, Ritsumeikan University, Kusatsu 525-8577, Japan

**Keywords:** N-confused porphyrin, cyclopropanation reaction, theoretical calculation, acceleration effect, *trans/cis* selectivity

## Abstract

The factors that affect acceleration and high *trans*/*cis* selectivity in the catalytic cyclopropanation reaction of styrene with ethyl diazoacetate by cobalt N-confused porphyrin (NCP) complexes were investigated using density functional theory calculations. The reaction rate was primarily related to the energy gap between the cobalt–carbene adduct intermediates, **A** and **B**, which was affected by the NCP skeletons and axial pyridine ligands more than the corresponding porphyrin complex. In addition, high *trans*/*cis* stereoselectivity was determined at the **TS1** and, in part, in the isomerization process at the carbon-centered radical intermediates, **C*_trans_*** and **C*_ci_******_s_***.

## 1. Introduction

Metalloporphyrins can catalyze fundamental reactions in biological systems and laboratories [1,2]. Various catalytic atom transfer reactions, including carbene, nitrene, and oxygen atoms [3,4,5], have been achieved by synthetic metalloporphyrins. Among such metalloporphyrin-catalyzed reactions, carbene transfer to olefins has been developed to synthesize cyclopropane rings, essential structural motifs found in biologically active natural products and drug molecules [6,7]. So far, a variety of metalloporphyrin complexes (Rh [8,9], Os [10], Fe [11,12], Ru [10,13,14], Ir [15], Co [16,17], etc.) have been demonstrated to be effective for catalytic cyclopropanation reactions. Recently, inspired by the pentacoordinate heme structure [18], we designed and synthesized a series of N-confused porphyrin (NCP) complexes having tethered axial ligands using the reactions of inner-core carbon and 2-heteropyridines (Figure 1c) [19,20]. When the cobalt–NCP complexes (e.g., the reduced form of **1a**) were subjected to the cyclopropanation reaction of styrene with ethyl diazoacetate (EDA), enhanced catalytic activity in both the reaction rate and stereoselectivity was observed, compared with the corresponding regular porphyrin. Thus, the structure–activity relationship with these porphyrin complexes remained a question to be solved.

NCP [21,22] is a porphyrin analog or, more precisely, a porphyrin isomer with an inverted pyrrole ring, which provides an NNNC core for metal coordination [23]. Due to the presence of the carbon atom inside, the introduction of a fifth coordinating ligand becomes facile. In addition, peculiar NH tautomerism between the inside and periphery makes NCP a unique chameleon-type ligand exhibiting different types of NH tautomeric forms (**2H**- and **3H**-form according to the number of hydrogen atoms in the core) (Figure 1a) [24]. The trianionic nature of the **3H**- (and **3H′**-) form and the σ-donating effect of the inner carbon atom through the carbon–metal bond allow for the stabilization of the high-valent metal ions in the NCP core [25,26]. Due to the intriguing coordination capabilities, NCP metal complexes have been subjected to several catalytic reactions such as cyclopropanation by rhodium and cobalt complexes [27,28], oxygen atom transfer from pyridine *N*-oxide by rhenium oxo complexes [29], styrene oxidation with PhIO by manganese complex [30], and cyclic carbonate formation between epoxide and carbon dioxide by nickel, palladium, and zinc complexes [31,32].

In particular, the cyclopropanation reaction of styrene with EDA using rhodium(III, IV)–NCP catalysts showed notable catalytic activity in yield and *trans*/*cis* (*t*/*c*) selectivity compared with that of regular porphyrin (e.g., 92% vs. 71%; *t*/*c* = 91/9 vs. 52/48) [27]. Fields et al. reported a similar high selectivity of cobalt(II) *N*-methylated N-confused tetraphenylporphyrin having an axial pyridine ligand, [**Co(MeNCTPP)(py)**] (e.g., 85% vs. 67%; *t*/*c* = 93/7 vs. 74/26) (Table 1, Entries 1–2) [28]. When we tried the same cyclopropanation reactions with 0.5 mol% of reduced **1a** (obtained by treating with aqueous sodium hydrosulfite), the reaction was completed within 5 min at room temperature, resulting in 78% yield and 92/8 of *trans*/*cis* selectivity [19]. Compared with the reported **Co(MeNCTPP)(py)** and the reference cobalt(II) tetraphenylporphyrin complex [**Co(TPP)**], the catalytic reactivity was significantly enhanced (Table 1, Entries 4–5). **Co(MeNCTPP)(py)** catalyzed the cyclopropanation reaction similar to **Co(NCTPPSpy)** at room temperature but with less than half of the turnover number (Table 1, Entries 3–4). On the other hand, **Co(TPP)** was almost inactive for this cyclopropanation reaction at room temperature. Aiming to clarify the origin of the enhanced catalytic ability of **Co(NCTPPSpy)**, we herein conducted density functional theory (DFT) calculations referring to the proposed mechanism of the cobalt(II) porphyrin-catalyzed cyclopropanation reaction by Zhang and de Bruin [33,34]. In the calculations, we focus on the factors that affect the reaction rate and the *trans*/*cis* selectivity of **Co(NCTPPSpy)** and **Co(MeNCTPP)(py)** compared with **Co(TPP)**.

## 2. Results and Discussion

### 2.1. Reaction Mechanism

The catalysts used for DFT calculations were **Co(TPP)**, **Co(MeNCTPP)**, **Co(MeNCTPP)(py)** and **Co(NCTPPSpy)** (Table 1). **Co(MeNCTPP)** has the same NCP skeleton as **Co(MeNCTPP)(py)** without an axial pyridine ligand, and **Co(NCTPPSpy)** is the supposed structure of the reduced form of **1a**. The mechanism adopted for calculating cyclopropanation reactions of styrene with EDA is shown in Figure 1 [33]. The bridging carbene **A** (**A1** or **A2**) derived from EDA, and cobalt complex is assumed in equilibrium with radical carbene **B** (Figure 1, upper). Styrene contacts **B** to form a C–C bond, yielding γ-alkyl radical **C**. **C** comprises two states, **C*_trans_*** and **C*_cis_***, leading to the *trans* and *cis* configurations of the cyclopropane, respectively. **C*_trans_*** and **C*_cis_*** interconvert mutually by the C–C bond rotation. After the cleavage of the Co–C bond in **C**, the product complex (**PC**), comprising Co porphyrin and cyclopropane derivatives (*trans* and *cis*), is formed (Figure 1, bottom).

### 2.2. Energy Diagram for the Reaction with Co(NCTPPSpy)

The energy diagrams of the *trans* and *cis* products for the catalytic cyclopropanation reaction with various cobalt porphyrins were obtained by DFT calculations. As a representative, the energy diagram with **Co(NCTPPSpy)** is shown in Figure 2a. The diagrams with other catalysts are shown in Appendix A, and the relative energies of each state are summarized in Appendix A. In all cases, the reaction pathways involve intermediates (**C*_trans_*** and **C*_cis_***) and products (**PC*_trans_*** and **PC*_cis_***). In the first step, the cyclopropanation reaction initiates with the C–N bond dissociation of **A2** (**A1** is less stable than **A2** by 1.1 kcal/mol, Appendix A), resulting in the formation of radical carbene **B** in the doublet state. (Note: the quartet and sextet states are in higher energy by 28.7 and 35.8 kcal/mol, respectively.) The rotation isomer **B’** has a higher energy of 0.6 kcal/mol than **B** with a rotation barrier (45.1*i* cm^−1^) of 4.3 kcal/mol. The dihedral angles (inner C–Co-radical C–carbonyl C) of **B**, **B’** and a transition state are 126.7°, 231.0°, and 174.9°, respectively. Since the energy difference between **A2** and **B** is small (0.4 kcal/mol), we expect that **B** plays the role of a possible near-attack conformation in the cyclopropanation reaction. The bond formation between **B** and styrene occurs via **TS1** with activation energies of 22.3 and 22.9 kcal/mol for **TS1*_trans_*** and **TS1*_cis_***, leading to **C*_trans_*** and **C*_cis_***, respectively (Figure 1, bottom). An intramolecular radical-radical coupling succeedingly occurs to form the cyclopropane complex **PC** via **TS2**. The latter reaction proceeds smoothly because of the small activation energies of 2.4 kcal/mol in **TS2*_trans_*** and 2.0 kcal/mol in **TS2*_cis_***. Computed results conclude that the rate-determining step of this cyclopropanation reaction is the C–C bond formation between the radical carbene **B** and styrene, namely at **TS1**, consistent with the reported studies [33,34].

The *trans*/*cis* selectivity of cyclopropanation reaction is attributed to the **TS1** state in the literature [33]. However, the calculated energy difference (0.6 kcal/mol) between **TS1*_trans_*** and **TS1*_cis_*** seems to be small to quantitively explain the observed *trans*/*cis* selectivity (*t*/*c* = 92/8, Table 1), which corresponds to 2.3 kcal/mol. Thus, we investigated the possibility of an intervening isomerization pathway at the stage of γ-alkyl radical intermediates, **C*_cis_*** and **C*_trans_***, with an energy barrier of **TSi** (Figure 2b). When the energy barrier of the **TSi** is lower than that of **TS2**, the *trans/cis* isomerization could affect the stereoselectivity (Figure 2a). Conversely, when the height of **TS2** is lower than **TSi**, the *trans*/*cis* selectivity is irrelevant to the **TSi** (Figure 2b) because of fewer opportunities for interconversion. In this study, we focused on the steps from **A** to **TS1** and the energy barrier of **TSi** and **TS2*_cis_*** to explain the acceleration effect and the high *trans/cis* selectivity in the Co-NCP catalyzed cyclopropanation reactions.

The calculated energy differences between *cis* and *trans* forms of transition states and intermediates are much smaller than we expected for discussing the experimental results quantitatively. However, we could see some trends among the catalytic systems examined. In the following sections, we will discuss the factors affecting the reaction rate and *trans*/*cis* selectivity based on the calculation and the experimental results.

### 2.3. Acceleration Effect

The rate of our cyclopropanation reaction depends on the relative Gibbs free energy [Δ*G*(**B**–**A**)] and the activation energy for **TS1** from **B** [*E_a_*(**TS1**)] (Appendix A). After **TS1**, the reaction goes to the product exergonically. The calculated Δ*G*(**B**–**A**) and *E_a_*(**TS1*_trans_***) values for cobalt porphyrin catalysts are summarized in Table 2. *E_a_*(**TS1*_trans_***) of **Co(NCTPPSpy)** (22.3 kcal/mol) is higher than that of **Co(TPP)** (18.7 kcal/mol). However, the relative energy for **TS1*_trans_*** of **Co(NCTPPSpy)** (22.7 kcal/mol) from the **A** state is lower than that of **Co(TPP)** (23.5 kcal/mol) due to the contribution of Δ*G*(**B**–**A**). The cyclopropanation reaction with **Co(NCTPPSpy)** catalyst proceeded smoothly at room temperature, whereas **Co(TPP)** did not show catalytic activity under the same conditions. Therefore, the reaction rates are likely dependent on Δ*G*(**B**–**A**) rather than *E_a_*(**TS1**), and the smaller Δ*G*(**B**–**A**) may accelerate the reaction. A similar trend was observed with other NCP catalysts. Because **A** and **B** are in an equilibrium state (**A** ⇄ **B**), a smaller Δ*G*(**B**–**A**) increases the opportunity of the reaction between **B** and styrene, which results in the acceleration of the cyclopropanation reaction.

### 2.4. Structures of ***A*** and ***B***

To clarify the factors that afford the much smaller Δ*G*(**B**–**A**) values for **Co(MeNCTPP)(py)** and **Co(NCTPPSpy)** than those for **Co(TPP)** and **Co(MeNCTPP)**, the structures of **A** and **B** states are analyzed in detail. For **Co(TPP)** and **Co(MeNCTPP)**, the changes in the bond length between the cobalt and the carbene atom in **A** and **B** are around 0.1 Å (0.091 and 0.122 Å, respectively). The formal valency of the cobalt center in **A** and **B** is different, divalent in **A** and trivalent in **B**. Owing to the σ-donation from the inner carbon atom of the NCP ligand, the trivalent **B** state of **Co(MeNCTPP)** is stabilized to decrease the energy gap between **A** and **B**.

On the other hand, for **Co(MeNCTPP)(py)** and **Co(NCTPPSpy)**, the corresponding bond-length difference in **A** and **B** is much smaller, less than 0.05 Å (0.047 and 0.032 Å). The valency of cobalt centers is trivalent in both **A** and **B**, and the spin density is delocalized on the NCP skeleton, different from **Co(TPP)** and **Co(MeNCTPP)**. Trivalent cobalt centers in both states are stabilized by NCP skeletons and the axial pyridine ligands. Consequently, the energy gap between **A** and **B** becomes smaller. Stabilizing the trivalent cobalt center by the NCP skeletons and the electron donation from the pyridine ligands are essential factors for accelerating the catalytic cyclopropanation reaction of styrene.

### 2.5. Trans/Cis Stereoselectivity

In the energy diagram with **Co(NCTPPSpy)**, the activation energies from **C*_cis_*** to **C*_trans_*** via **TSi**, *E_a_*[**TSi**(**C*_cis_***→**C*_trans_***)], and **TS2**, *E_a_*(**TS2*_cis_***), were significantly lower than *E_a_*(**TS1*_trans_*_/*cis*_**) (Figure 2). This result suggests that **C** formed by passing through **TS1** is rapidly converted to **PC** via **TS2**. However, the possibility of conversion of **C*_cis_*** to **C*_trans_*** via **TSi** remains because *E_a_*[**TSi**(**C*_cis_***→**C*_trans_***)] is lower than *E_a_*(**TS2*_cis_***) by 0.5 kcal/mol, and **C*_trans_*** is more stable than **C*_cis_*** by 0.4 kcal/mol. We assume this bypath process is kinetically controlled, and the formed **C*_trans_*** goes to **PC** via **TS2*_trans_*** without reaching the equilibrium between **C*_cis_*** to **C*_trans_***. Although the reverse process, **C*_trans_***→**C*_cis_*** via **TSi**, cannot be ignored because of a small energy barrier difference (0.4 kcal/mol), it might be energetically less favorable. This situation is manifested in the **Co(MeNCTPP)(py)** case, which shows energy barriers of 3.3 kcal/mol vs. 1.9 kcal/mol, for **C*_trans_***→**C*_cis_*** and **C*_cis_***→**C*_trans_***, respectively (Appendix A). To summarize, a large portion of **C** goes to **PC** via **TS2** without interconversion. In addition to this pathway, a small part of **C*_cis_*** isomerizes into more stable **C*_trans_*** via **TSi**, increasing the population of **C*_trans_***. Consequently, **Co(NCTPPSpy)** and **Co(MeNCTPP)(py)** exhibit high *trans* selectivity in the catalytic system. These results are attributed to the energy difference between **TS1*_trans_*** and **TS1*_cis_*** and the isomerization from **C*_cis_*** to **C*_trans_*** via **TSi** (Figure 2a). On the other hand, *E_a_*[**TSi**(**C*_cis_***→**C*_trans_***)] is more significant than *E_a_*(**TS2*_cis_***) by 2.7 kcal/mol for **Co(TPP)**, which loses the opportunity of conversion into **C*_trans_*** via **TSi** (Figure 2b). Although the energy differences we used in the discussion are too small to correctly evaluate the *trans/cis* ratios at our calculation level, the selectivity trends agree with the experimental results. Thus, we think that a further high *trans*/*cis* selectivity could be realized by controlling *E_a_*(**TSi**) and *E_a_*(**TS2**) with suitable substituents in **Co(NCTPPSpy)** and **Co(MeNCTPP)(py)**.

### 2.6. Structures of ***C***

Regarding the higher **TS2** levels of NCP catalysts than the regular porphyrin, the structures of γ-alkyl radical intermediate **C** and **TS2** were compared in detail. The bond lengths between the cobalt and carbene atoms are similar in all **C**, [**Co(TPP)**; **C*_trans_*** (**C*_cis_***): 2.067 (2.060) Å, **Co(MeNCTPP)**; 2.030 (2.030) Å, **Co(MeNCTPP)(py)**; 2.062 (2.062) Å, **Co(NCTPPSpy)**; 2.076 (2.073) Å]. In contrast, those values in **TS2** are largely affected by the NCP skeletons and pyridine ligands: NCP complexes show longer bond lengths than that of **Co(TPP)**, [**Co(TPP)**; **TS2*_trans_*** (**TS2*_cis_***): 2.180 (2.194) Å, **Co(MeNCTPP)**; 2.289 (2.299) Å, **Co(MeNCTPP)(py)**; 2.382 (2.397) Å, **Co(NCTPPSpy)**; 2.408 (2.401) Å]. For **Co(NCTPPSpy)**, the pyridine nitrogen atom strongly contacts the cobalt center compared with **Co(MeNCTPP)(py)**. Due to the electron donation from the axial pyridine ligand to the cobalt center, the structural changes become larger in **TS2** for the pyridine-coordinating catalysts. Consequently, the reactions with **Co(NCTPPSpy)** and **Co(MeNCTPP)(py)** go through both the **TS2** and **TSi** to exhibit high *trans*/*cis* selectivity.

## 3. Conclusions

The factors affecting acceleration and *trans*/*cis* selectivity in the catalytic cyclopropanation reaction of styrene with ethyl diazoacetate by cobalt N-confused porphyrin (NCP) complexes were investigated with DFT calculations in this work. The reaction rates are affected by the energy gap (Δ*G*(**B**–**A**)) between the cobalt–carbene adduct intermediates, **A** and **B**, which is largely decreased by the NCP skeletons and axial pyridine ligands. **Co(MeNCTPP)(py)** and **Co(NCTPPSpy)** exhibit smaller Δ*G*(**B**–**A**) values than that of **Co(TPP)**, leading to higher reaction rates than **Co(TPP)**, as illustrated experimentally. High *trans*/*cis* selectivity originates from the energy difference in **TS1** and, in addition, the isomerization of γ-alkyl radical intermediate **C**. For **Co(TPP)**, the conversion from **C*_cis_*** to **C*_trans_*** is negligible, whereas **Co(MeNCTPP)(py)** and **Co(NCTPPSpy)** can take a bypath route to increase the population of **C*_trans_***. Consequently, cyclopropanation products with high *trans*/*cis* selectivity were achieved. 

This study revealed that minute changes in the porphyrin structure, such as N-confusion, cause a distinct difference in the energy level of intermediates and pathways in the catalytic reaction. Furthermore, because of the structural resemblance to regular porphyrin, NCP metal complexes could also be applied for the cofactor-replacing modification of biocatalysts [35,36]. The research in this direction is currently underway in our laboratory.

## 4. Materials and Methods

Calculation methods: The DFT calculations were conducted with the Gaussian 09 program package (Rev. E.01) [37]. We used the B3LYP functional [38,39,40] combined with the (15s11p6d) primitive set of Wachters–Hay supplemented with one polarization f-function (α = 1.17) [41,42,43] for cobalt atoms and the D95** basis set [44] for the other atoms. After geometry optimizations, vibrational analyses were performed for all reaction species to confirm stable and transition structures. The spin multiplicity and electronic charge analysis were performed in the doublet state (S = 1/2) and neutral in all calculations, respectively. Energy profiles of calculated pathways are presented as Gibbs free energy changes (in kcal/mol) at 298.15 K.

## Data Availability

Not applicable.

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
