# Peer review of "A Computational Study on the Mechanism of Catalytic Cyclopropanation Reaction with Cobalt N-Confused Porphyrin: The Effects of Inner Carbon and Intramolecular Axial Ligand"

_molecules, 2022, doi:10.3390/molecules27217266_

Round 1
Reviewer 1 Report
The authors provide their computational study that rationalizes the effect of cobalt N-confused porphyrin complexes on the rate and selectivity of cyclopropanation reactions. The studies appear to be carried out carefully and, for the most part, provide good insight as to the effect of porphyrin structure on catalyst reactivity. I do have a few suggestions that may improve the manuscript:
1. The experimental reactions are carried out in toluene as solvent while the calculations are carried out in vacuo. The differences in energies of the transition states and intermediates are fairly small. Did the authors investigate whether carrying out the calculations that incorporate solvent effects have an impact on the results? Is it possible that solvent effects alone, as subtle as they may be, might also contribute to the enhanced rates/specificities?
2. Page 4, line 111: the rotational isomer B' is mentioned in the text, but there is not structure available for viewing. This should probably be included.
3. Page 6, lines 200-202: This sentence seems to be incomplete. Perhaps "... the selectivity trends are much aligned with the experimental results."
Otherwise, the paper is clearly written and does bring insight as to the effect of the catalyst on the cyclopropanation reaction. With attention to the above suggestions, I would support publication.
Reviewer 2 Report
The article describes computational analysis of the cyclopropanation reaction of styrene with ethyl diazoacetate by catalyzed by cobalt complexes with N-confused porphyrin ligands. The authors addressed an interesting question of the origin of improved catalytic activity and selectivity of the complexes with NCPs. They completed substantial amount of computations of intermediates and TSs of the reactions catalyzed by four complexes. The calculations seem sound, and comparison of the energies of intermediates are reasonable. However, they don’t really support the authors conclusions.
The first problem is the fact that the differences in energies of different intermediates and TSs are small, mostly less than 1 kcal/mol. In general, the errors in reproducing experimental values of DG and E(TS) by computations are larger, especially when radical species and/or complexes with variable oxidations states are involved. In this work, estimation of the rates of reactions using calculated DG via eq 4 from SI produces very small values 10-7 M s-1. This is probably several order of magnitude less than that obtained in experiments. The estimation of the relative energies of intermediates and TSs is probably better. But even if the differences in energies are more accurate than the absolute values, they still do not explain differences in selectivity and reaction rates.
In particular, the authors correctly derived expression for rates shown in eq 4 in SI, It indicates that the effective barriers for the reactions are determined by the sum Δ?(? − ?) + ?a(TS1). So, Δ?(? − ?) does affect rates, but it does not control them, as author concluded, because larger Δ?(? − ?) in reaction involving Co(TPP) is compensated by the lower Ea(TS1) in these systems. So, overall differences in rates are determined by Δ?(? − ?) + ?a(TS1) which are quite small.
Second, the authors concluded that in systems with E(TS2)> E(TSi), some of cis isomers would convert to trans isomers. Such differences in TS indeed suggest that interconversions occur. However, negative values of DG(Ctrans–Ccis) indicate that cis would convert to trans isomers only if the starting concentrations of both were the same. If TS1-determined concentrations of cis isomers are smaller than the equilibrium concentrations determined by DG(Ctrans–Ccis), the opposite process (trans to cis) would occur. This could make concentration of the trans isomers smaller than that determined by the TS1. Specifically, Co(NCTPPSpy) with very small G(Ctrans–Ccis) of -0.4 kcal/mol should be least selective. This contradict the data in Table 1,So, high trans/cis stereoselectivity cannot be related to the interconversion process between the carbon-centered radical intermediates, Ctrans and Ccis .
Since the Abstract states these two incorrect or at least misleading conclusions as the main findings of the paper, it cannot be recommended for publication.
Reviewer 3 Report
Manuscript by Furuta et al. reports on factors controlling acceleration and trans/cis selectivity in the catalytic cyclopropanation reaction of styrene with ethyl diazoacetate by cobalt N-confused porphyrin complexes investigated with DFT methods. Furuta is a leading expert in the field of N-confused porphyrins (isomers of porphyrins) and also a discoverer of these types of porphyrins. In their previous experimental work, authors observed higher activity of N-confused porphyrin metal complexes vs. regular porphyrin complexes. This contribution describes QM findings regarding possible mechanisms of the catalytic reaction, intermediate states and also potential transition states. Results thus obtained are in accordance with the experimental data and will help researchers to design novel more effective catalysts. I strongly support the acceptance of this contribution.
Round 2
Reviewer 2 Report
Although I still have reservation regarding analysis of kinetics and Ccis -Ctrans equilibria, the calculated energies are sound. Thus, with adjustments made by the authors, the article can be published.